# Host Lifeform Shapes Phyllospheric Microbiome Assembly in Mountain Lake: Deterministic Selection and Stochastic Colonization Dynamics

**DOI:** 10.3390/microorganisms13050960

**Published:** 2025-04-23

**Authors:** Qishan Xue, Jinxian Liu, Yirui Cao, Yuqi Wei

**Affiliations:** 1College of Environment and Resources, Shanxi University, Taiyuan 030006, China; xueqishan@sxdx.wecom.work (Q.X.); caoyirui@sxdx.wecom.work (Y.C.); weiyuqi@sxu.edu.cn (Y.W.); 2Shanxi Key Laboratory of Ecological Restoration on the Loess Plateau, Institute of Loess Plateau, Shanxi University, Taiyuan 030006, China

**Keywords:** epiphytic bacteria, diversity, assembly mechanisms, macrophyte, subalpine lakes

## Abstract

The phyllosphere microbiome of aquatic macrophytes constitutes an integral component of freshwater ecosystems, serving crucial functions in global biogeochemical cycling and anthropogenic pollutant remediation. In this study, we examined the assembly mechanisms of epiphytic bacterial communities across four phylogenetically diverse macrophyte species (*Scirpus validus*, *Hippuris vulgaris*, *Nymphoides peltatum*, and *Myriophyllum spicatum*) inhabiting Ningwu Mayinghai Lake (38.87° N, 112.20° E), a vulnerable subalpine freshwater system in Shanxi Province, China. Through 16S rRNA amplicon sequencing, we demonstrate marked phyllospheric microbiome divergence, as follows: Gammaproteobacteria dominated *S. validus*, *H. vulgaris* and *N. peltatum*, while Alphaproteobacteria dominated in *M. spicatum*. The nitrate, nitrite, and pH value of water bodies and the chlorophyll, leaf nitrogen, and carbon contents of plant leaves are the main driving forces affecting the changes in the β-diversity of epiphytic bacterial communities of four plant species. The partitioning of assembly processes revealed that deterministic dominance governed *S. validus* and *M. spicatum*, where niche-based selection contributed 67.5% and 100% to community assembly, respectively. Conversely, stochastic processes explained 100% of the variability in *H. vulgaris* and *N. peltatum* microbiomes, predominantly mediated by dispersal limitation and ecological drift. This investigation advances the understanding of microbial community structural dynamics and diversity stabilization strategies in aquatic macrophyte-associated microbiomes, while establishing conceptual frameworks between plant–microbe symbiosis and the ecological homeostasis mechanisms within vulnerable subalpine freshwater ecosystems. The empirical references derived from these findings offer novel perspectives for developing conservation strategies aimed at sustaining biodiversity equilibrium in high-altitude lake habitats, particularly in the climatically sensitive regions of north-central China.

## 1. Introduction

Aquatic plants are critical components of lake ecosystems, playing essential roles in maintaining ecosystem homeostasis, enhancing species diversity, mediating nitrogen and phosphorus biogeochemical cycles, and regulating ecological health [1,2]. Different types of aquatic plants exhibit distinct growth morphologies and distribution patterns within lakes [3]. Emergent plants, which have leaves extending above the water’s surface, form the upper structural layer of lake ecosystems, floating-leaved plants bridge the interface between water and air, and submerged plants establish the benthic structural layer. This multi-layered ecological architecture provides diverse niches for organisms, thereby increasing the biodiversity of lake ecosystems. Microbial communities attached to aquatic plant surfaces are pivotal drivers of material cycling in these systems [4]. These microorganisms include symbiotic and parasitic groups. Microorganisms act as key agents in nutrient cycling, as follows: Bacteria convert organic nitrogen into inorganic forms (e.g., decomposing proteins into ammonium salts, which are subsequently absorbed by plants). In phosphorus cycling, microbes degrade organic phosphorus compounds, releasing soluble phosphates to facilitate phosphorus exchange among water, plants, and microorganisms.

The distribution of epiphytic microbial communities on aquatic plants is influenced by multiple factors [5]. First, the physicochemical properties of surrounding water—such as temperature, pH, dissolved oxygen, and nutrient concentrations—affect microbial communities’ composition and abundance [6,7]. Second, host-specific traits, including plant species, ecological status, and surface morphology, regulate microbial colonization and growth [8,9]. At the same time, the mobility of water restricts microbial dispersal, leading to host specificity in epiphytic communities. The interactions between epiphytic microorganisms on plant leaf surfaces and their influencing factors differ significantly in soil and aquatic environments, which primarily manifests in ecological functions, community structures, and environmental-driving mechanisms. Terrestrial plant–leaf-associated microorganisms often form symbiotic relationships with plants. For instance, nitrogen-fixing bacteria (e.g., *Beijerinckia*) collaborate with plants for nitrogen fixation [1,10]. Because of the higher stability of soil environments, microbial community structures remain relatively stable. The composition of epiphytic microorganisms is directly regulated by secretions from different plants (e.g., legumes attract nitrogen-fixing bacteria, while grasses favor cellulose-degrading bacteria). Soil pH, organic matter content, and nutrient ratios (e.g., N:P) significantly influence microbial communities [11]. Whether in soil or water, the epiphytic microbial community of plant is subject to the selection of the owner. However, the composition, diversity and primary drivers of epiphytic bacterial communities across aquatic plant types remain understudied. The assembly of these communities into structured and functional groups results from the interplay of the following two fundamental processes: (1) deterministic processes, emphasizing environmental filtering that selects for specific taxa [12] and (2) stochastic processes, driven by dispersal limitations or ecological drift [13].

Although progress has been made in understanding the assembly mechanisms of plant-associated microbial communities [14], research has predominantly focused on terrestrial plants. Investigations into epiphytic bacterial communities on aquatic plants, particularly how plant lifeforms (e.g., emergent, floating-leaved, and submerged) influence community assembly, remain nascent [15]. In order to bridge these gaps, this study investigated the subalpine natural lakes in Ningwu County, Shanxi Province. Four major aquatic plant species in the subalpine lakes of Ningwu County, Shanxi Province, were selected as the research objects, namely, *Nymphoides peltatum*, *Myriophyllum spicatum*, *Hippuris vulgaris*, and *Scirpus validus*. *N. peltatum* (a floating-leaved perennial) thrives in sunlit and nutrient-rich waters. *M. spicatum* (a submerged perennial) predominantly inhabits nutrient-enriched environments with elevated nitrogen and/or phosphorus concentrations. *H. vulgaris* (an emergent perennial) mainly inhabits environments with sufficient light and muddy substrates at the bottom. *S. validus* (an emergent perennial) inhabits shallow water environments with abundant sunlight, high soil organic matter content, and loose soil texture. These four species, commonly found in wetlands and shallow lacustrine ecosystems across northern China, play crucial roles in sustaining aquatic biodiversity through habitat structuring and ecological niche partitioning. This study aims to clarify the impact of plant life forms on the structure of their bacterial communities and reveal the key driving factors controlling community assembly in different hosts by analyzing the composition, diversity, and assembly mechanisms of epiphytic bacterial communities of aquatic plants. This will deepen our understanding of the microbial ecological dynamics in lakes and provide a theoretical basis for the conservation of biodiversity and sustainable management of freshwater resources in subalpine regions.

## 2. Materials and Methods

### 2.1. Study Area and Sample Collection

Ningwu Mayinghai Lake (38.87° N, 112.20° E) is located at the foot of Guancen Mountain, 20 km southwest of Ningwu County, Shanxi Province, China. Situated on the eastern edge of the Loess Plateau, this subalpine lake is sensitive to East Asian monsoon dynamics. The lake has an average elevation of 1774 m, a surface area of 0.58 km^2^, an annual mean temperature of 6.2 °C, and an annual precipitation of 462.5 mm. Sampling was conducted in August 2020. Fresh leaves from four aquatic plants, including the emergent plant *Scirpus validus* (LSV), emergent plant *Hippuris vulgaris* (LHV), floating-leaved plant *Nymphoides peltatum* (LNP), and submerged plant *Myriophyllum spicatum* (LMS) (Figure 1). were collected in triplicate from shallow areas (~1 m from the shoreline). Samples were immediately placed into sterile Zip-lock bags. Concurrently, triplicate water samples near each plant species were collected in 1 L sterile plastic bottles for physicochemical analysis. All samples were transported to the laboratory in a 4 °C vehicle refrigerator and stored at −4 °C. The epiphytic bacterial communities and physicochemical properties of plants/water were analyzed within one week to ensure sample freshness.

Epiphytic bacterial sampling protocol: Leaf samples were weighed and transferred to sterile centrifuge tubes. A total of 0.1 mol L^−1^ potassium phosphate buffer (pH 8.0, pre-cooled to 4 °C) was added at a mass-to-volume ratio of 1 g:10 mL. Standardized washing was performed using an ultrasonic disruptor (200 W, 1 min) combined with vortexing (10 s), repeated twice to dislodge epiphytic microbes. Residual impurities were removed by repeating the procedure once. The combined wash solutions were vacuum filtered through 0.2 μm sterile membranes. The retained microbial biomass was sealed in cryotubes and stored at −20 °C for DNA extraction.

### 2.2. Experimental Procedures

#### 2.2.1. Physicochemical Parameter Analysis

Water parameters, including pH, electrical conductivity (EC), dissolved oxygen (DO), salinity (SAL), nitrate (NO_3_^−^), and ammonium (NH_4_^+^), were measured in situ using a portable multi-parameter water quality analyzer (Aquaread AP-5000, Broadstairs, UK). The total carbon (TC), total organic carbon (TOC), and inorganic carbon (IC) were quantified with a Shimadzu TOC analyzer (TOC-VCPH, Kyoto, Japan). Total nitrogen (TN), sulfate (SO_4_^2−^), and phosphate (PO_4_^3−^) concentrations were determined using an automated discrete chemical analyzer (Cleverchem Anna, DeChem-Tech, Hamburg, Germany). Leaf carbon (C), nitrogen (N), and sulfur (S) contents were measured with an elemental analyzer (Elementar Vario MACRO, Hannover, Germany). Chlorophyll, soluble protein, and soluble sugar levels in leaves were assayed using commercial kits (Sangon Biotech, Shanghai, China).

#### 2.2.2. DNA Extraction and High-Throughput Sequencing

Microbes retained on filters were eluted via vortexing in 1× phosphate-buffered saline (PBS). DNA was extracted using the Fast DNA SPIN Kit (MP Biomedicals, Santa Ana, CA, USA) following the manufacturer’s protocol. The DNA concentration and purity (A260/A280 ratio) were assessed via spectrophotometry. Qualified DNA was amplified targeting the V3–V4 hypervariable regions of bacterial 16S rDNA with primers 338F (5’-ACTCCTACGGGAGGCAGCA-3’) and 806R (5’-GGACTACHVGGGTWTCTAAT-3’), following Liu et al. [16]. Purified PCR products were sequenced on an Illumina MiSeq platform (Majorbio BioPharm Technology, Shanghai, China). The 16S rRNA gene raw sequence data were submitted to NCBI GenBank, with accession number: PRJNA1243075.

### 2.3. Data Analysis

Raw paired-end sequences were merged using FLASH, and chimeras were removed via the QIIME pipeline. High-quality sequences were clustered into operational taxonomic units (OTUs) at 97% similarity and annotated against the SILVA database (v138, confidence threshold: 70%). The OTU data were normalized to the minimum sequencing depth for diversity and community structure analyses. Differences in physicochemical parameters and α-diversity indices (Shannon, Simpson) among groups were evaluated using one-way ANOVA with Waller-Duncan post hoc tests in SPSS 22.0 (IBM, New York, NY, USA). Venn diagrams visualized unique and shared OTUs. Hierarchical clustering based on Bray–Curtis distances revealed significant beta diversity differences. Bray–Curtis distance-based hierarchical clustering was employed to assess spatial distribution patterns of bacterial communities. Environmental variables with variance inflation factors (VIFs) > 10 were excluded via multicollinearity tests. Mantel tests (“vegan” package in R) quantified the Spearman correlations between bacterial community composition and environmental/plant traits. The relative contributions of deterministic (selection and dispersal) and stochastic (drift and speciation) processes to community assembly were assessed using the βNTI framework [17] implemented in the “iCAMP” R package. A path analysis (“lavaan” package) compared the effects of water chemistry and leaf traits on bacterial communities. Statistical significance was set at *p* < 0.05 (95% confidence level).

## 3. Results

### 3.1. Water Physicochemical Properties and Leaf Physiological Traits

Among 13 water quality parameters, DO, TN, and PO_4_^3−^ differed significantly across plant species (*p* < 0.05; Table 1). DO was highest in the water surrounding *Nymphoides peltatum*; NH_4_^+^ peaked near *Myriophyllum spicatum*; and carbon fractions (TC, TOC, and IC) were highest near *Scirpus validus*. The pH decreased near *Hippuris vulgaris* and *N. peltatum*, while SAL and SO_4_^2−^ declined near *N. peltatum*, *M. spicatum* and *S. validus*, *H. vulgaris*, respectively (*p* < 0.05). Leaf carbon and nitrogen contents varied significantly among species (*p* < 0.05), whereas the sulfur content showed no difference. The soluble protein was lowest in *M. spicatum* leaves, chlorophyll content was highest in *H. vulgaris*, and soluble sugars were elevated in *S. validus* and *N. peltatum* compared to other species (*p* < 0.05; Table 2).

### 3.2. Species Composition of Bacterial Communities

Epiphytic bacterial communities exhibited distinct compositional differences among aquatic plant species at both the phylum and genus levels. High-throughput sequencing of 12 samples yielded a total of 3580 operational taxonomic units (OTUs), spanning 145 classes, 341 orders, 612 families, 1041 genera, and 2043 species. At the phylum level, Proteobacteria, Firmicutes, Bacteroidota, Actinobacteriota, and Cyanobacteria were shared across all four plant leaves but displayed significant variations in relative abundance (Figure 2A). Proteobacteria dominated as the most abundant phylum on leaves of *H. vulgaris* (72.3%), *M. spicatum* (47.7%), *N. peltatum* (49.5%), and *S. validus* (53.2%). At the genus level, 40 dominant genera (relative abundance > 1%) were identified across all samples (Figure 2B). *Pseudomonas* was the most abundant genus on *H. vulgaris* (66%), while *Exiguobacterium* dominated *M. spicatum* (12.4%) and *S. validus* (19.8%). *Flavobacterium* showed the highest relative abundance on *N. peltatum*. At the class level, epiphytic bacterial communities revealed differences in the composition of the bacterial communities on the leaves of different plants. Alphaproteobacteria demonstrated the highest relative abundance on *M. spicatum*, while Gammaproteobacteria dominated the phyllosphere communities of the remaining three plant species (Appendix A).

At the OTU level, the epiphytic bacterial communities on leaves of *H. vulgaris* (LHV), *M. spicatum* (LMS), *N. peltatum* (LNP), and *S. validus* (LSV) comprised 273, 1096, 1085, and 995 OTUs, respectively. The four plant species shared 24 common OTUs (Figure 3A), including 12 dominant shared OTUs (relative abundance > 1%, Figure 3B) affiliated with three bacterial phyla, as follows: Proteobacteria (9 OTUs), Actinobacteriota (2 OTUs), and Firmicutes (1 OTU).

In addition, the most abundant unique OTUs for each plant were OTU357 (unclassified *Pseudarthrobacter*) on LHV, OTU35 (unclassified *Pseudomonas*) on LMS, OTU1809 (unclassified *Mycobacterium*) on LNP, and OTU1701 (*Acinetobacter lwoffii*) on LSV (Figure 3B).

### 3.3. Alpha Diversity of Bacterial Communities

Alpha diversity indices (richness, Shannon, and Simpson) revealed significant differences among communities. *H. vulgaris* (LHV) exhibited the lowest diversity (richness = 107.14, Shannon = 0.80, and Simpson = 0.325), which was significantly lower than *M. spicatum* (LMS) and *N. peltatum* (LNP) (*p* < 0.05). In contrast, no significant difference was detected between LHV and *S. validus* (LSV) in terms of these diversity indices. Moreover, the LMS community displayed a relatively high richness, indicative of a greater number of species present. The Shannon indexes for LMS and LNP were moderately high, suggesting a relatively even distribution of species within these communities. For the Simpson index, LNP showed a value that was higher than those for LMS and LSV, reflecting a different pattern of dominance and diversity within the community (Figure 4).

### 3.4. Spatial Distribution Patterns and Driving Factors of Bacterial Communities

Hierarchical clustering based on Bray–Curtis distances at the OTU level grouped the four plant-associated bacterial communities into two clusters (Figure 5). The emergent plants (i.e., LHV and LSV) formed one cluster, while the submerged (i.e., LMS) and floating-leaved (i.e., LNP) plants formed another, indicating higher structural similarity within the lifeforms.

In this study, Pearson correlation analysis was employed to investigate how the physical and chemical parameters of water bodies and leaf physiological attributes of plants influence the construction of the epiphytic bacterial communities. Mantel tests results revealed that many factors had significant impacts on the epiphytic bacterial communities. Notably, the nitrogen concentration in the water body had an extremely significant influence, including TN, ${NH}_{4}^{+}$, C/N, and N_P_. Additionally, pH, TC, IC, C_P_, and sugar also had significant effects on the epiphytic bacterial communities (Figure 6).

### 3.5. Assembly Processes and Influencing Factors of Epiphytic Bacterial Communities

The null model analysis based on βNTI values demonstrated that there were no significant differences between the pairs of communities LNP and LHV, as well as LHV and LSV (*p* > 0.05). In contrast, significant differences were observed among the remaining communities (*p* < 0.05). A further combined analysis of βNTI and RCBray revealed that stochastic processes were dominant in the LNP and LHV communities. Conversely, deterministic processes governed the community assembly in the LSV and LMS communities (Figure 7).

A path analysis demonstrated that bacterial communities on floating-leaved (LNP) and submerged (LMS) plants were primarily shaped by the leaf chlorophyll a (Chla), chlorophyll b (Chlb), nitrogen (N), and carbon (C) contents. In contrast, communities on emergent plants (LSV and LHV) were driven by water chemistry, including SO_4_^2−^, NO_2_^−^, NO_3_^−^, and pH (Figure 8).

## 4. Discussion

### 4.1. Plant Life Forms and Assembly of Epiphytic Bacterial Communities

#### 4.1.1. Differences in the Species Compositions of Bacterial Communities

Species compositional diversity is not only a critical determinant of community properties but also a core feature for distinguishing between community types [18]. Due to the different response patterns of the various microbial groups to environmental and spatial variables, there may be differences in the community assembly processes of species with different relative abundances and occurrence frequencies. In this study, we analyzed the composition and diversity of epiphytic bacterial communities on the leaves of four aquatic plants: *Scirpus validus*, *Hippuris vulgaris*, *Myriophyllum spicatum*, and *N. peltatum*. The results revealed that Proteobacteria was the most abundant phylum on the leaves of *S. validus*, *H. vulgaris*, *M. spicatum*, and *N. peltatum*. consistent with prior studies highlighting its dominance in biogeochemical cycling [19]. At the class level, Gammaproteobacteria dominated *S. validus* and *H. vulgaris*, whereas Alphaproteobacteria prevailed on *M. spicatum* (Figure 2A). For *N. peltatum*, *Mycobacterium* was the most abundant genus. The high metabolic versatility of *Pseudomonas* (dominant on *H. vulgaris*), including extracellular polysaccharide production critical for biofilm formation [20,21], underscores the role of host-specific carbon sources in shaping bacterial communities [22]. These findings demonstrate that distinct microhabitat conditions on different plant leaves drive significant compositional divergence in epiphytic bacteria, providing new insights into plant–microbe interactions in aquatic ecosystems.

#### 4.1.2. Diversity Patterns and Driving Factors

The epiphytic bacterial diversity varied markedly across plant species. The migration and growth of these bacteria are non-random and constrained by host traits and environmental conditions [23]. Submerged *M. spicatum* exhibited the highest alpha diversity, followed by floating-leaved *N. peltatum*, while emergent *S. validus* and *H. vulgaris* showed lower diversity (Figure 4). This gradient likely reflects environmental stability, as follows: submerged plants, which are fully immersed in water, develop stable biofilms that support diverse microbial niches [24]. In contrast, emergent plants experience fluctuating conditions (e.g., air exposure and water level changes), disrupting biofilm integrity [25]. Morphological differences also contribute; for instance, the filamentous leaves of *M. spicatum* provide greater surface area for bacterial colonization compared to the linear, smooth leaves of *S. validus* and *H. vulgaris* [26,27].

Hierarchical clustering based on Bray–Curtis distances revealed significant beta diversity differences (Figure 5). Communities on emergent plants (*H. vulgaris* and *S. validus*) clustered separately from those on submerged (*M. spicatum*) and floating-leaved (*N. peltatum*) plants, indicating higher structural similarity within lifeforms [28]. Mantel tests identified water chemistry (NH_4_^+^, C/N ratio, TN, and SAL) and leaf nitrogen content as primary drivers of beta diversity (Figure 6). Leaf nitrogen, a proxy for photosynthetic capacity [29], influences bacterial diversity by modulating host-derived exudates. Water nitrogen levels further alter exudate composition, particularly favoring denitrifying bacteria in biofilms [30,31,32]. These findings highlight that both environmental filtering and host physiology jointly shape epiphytic community structure.

#### 4.1.3. Assembly Processes and Influencing Factors

The assembly mechanisms of plant-associated epiphytic bacterial communities, whether dominated by stochastic processes, represent a central question in microbial ecology. Although traditional perspectives emphasize host selection pressure (i.e., deterministic processes) as the key driver of epiphytic microbial community formation, our study suggests that stochastic processes may play significant roles under specific conditions. Stochastic processes dominated in *N. peltatum* (floating-leaved) and *H. vulgaris* (emergent), whereas deterministic processes (homogeneous selection) controlled *M. spicatum* (submerged) and *S. validus* (emergent) (Figure 7). The leaves of *N. peltatum* float on the water’s surface, while *H. vulgaris* exhibits a thalloid structure. Both species feature open epiphytic surfaces that accommodate microbial colonization with simple physical architectures and likely secrete limited antimicrobial compounds (e.g., phenolics and terpenoids), resulting in weak chemical barriers against microbial colonization. Additionally, their low production of metabolites, such as soluble sugars or amino acids, fails to impose stable metabolic selection pressures [33]. In addition, surface water currents may still drive passive microbial detachment (e.g., via leaf swaying), and the frequent resetting of communities leads to a stronger reliance on stochastic colonization events [34]. The host filtering effect and physicochemical factors of water bodies jointly shape the assembly process of epiphytic bacterial communities on *M. spicatum* and *S. validus*. The upright culms of *S. validus* secrete alkaloids that inhibit colonization by certain bacterial taxa [28], while their waxy-coated surfaces create hydrophobic microenvironments that selectively retain desiccation-tolerant [35] extracellular polysaccharide-producing bacteria. In contrast, *M. spicatum* releases sesquiterpene lactones that specifically enrich microorganisms possessing corresponding degradation genes [36]. Concurrently, photosynthetic oxygen released from its submerged stems and leaves forms oxidized microzones, driving deterministic selection of aerobic bacteria.

### 4.2. Research Implications

In the process of global ecological environment dynamics, lakes, as key components of ecosystems, respond extremely rapidly to environmental changes and can be regarded as frontier sentinels of environmental change response [37]. Among them, subalpine lakes, with their notable feature of being less intervened by human activities, have unique advantages in reflecting climate change.

Through our research, it has been found that the epiphytic bacterial community exhibits significant host specificity, and the host lifestyle plays a decisive role in the construction process of the epiphytic bacterial community. Notably, the denitrifying bacteria group has a relatively high abundance in the epiphytic bacterial community, which enables it to play a pivotal role in the treatment of lake eutrophication, especially in the degradation process of excessive nitrogen. Meanwhile, human activities, climate change [38], and the continuous reduction in lake ice cover area [39] have had a profound impact on the bacterial community structure, and the bacterial community’s composition will actively respond to dynamic changes in the environment. This response does not exist in isolation. It will further have a chain effect on the nitrogen and phosphorus cycles in lakes, thus intensifying the feedback of the lake ecosystem to climate change in turn and forming a complex coupling relationship.

Therefore, conducting systematic research on the epiphytic bacterial community of subalpine lakes is of far-reaching significance. As a sensitive indicator of ecological changes, it can not only enable us to capture climate change signals in a timely manner but also provide key theoretical support for the formulation and implementation of biological control strategies for climate change.

## 5. Conclusions

The assembly of epiphytic bacterial communities on aquatic macrophytes in a vulnerable subalpine lake ecosystem is governed by host lifeform acting as a hierarchical ecological filter. Distinct microbial divergence was observed across the following four macrophyte species: Gammaproteobacteria dominated *Scirpus validus* (emergent) and *Hippuris vulgaris* (emergent), *Flavobacterium* prevailed on *Nymphoides peltatum* (floating-leaved), and Alphaproteobacteria characterized *Myriophyllum spicatum* (submerged). Microbial β-diversity was primarily driven by water chemistry and plant functional traits. The assembly mechanisms differed significantly; deterministic processes (niche-based selection) accounted for 67.5% and 100% of the community assembly in *S. validus* and *M. spicatum*, respectively, reflecting environmental filtering and host-specific adaptations. In contrast, stochastic processes entirely governed the microbiomes of *H. vulgaris* and *N. peltatum*, highlighting the role of random colonization in stable aquatic niches. These findings underscore the critical influence of plant lifeforms in structuring phyllosphere microbiomes, with submerged plants favoring deterministic assembly linked to pollutant remediation and emergent-/floating-species hosting stochastic communities that may enhance ecosystem resilience. By revealing how host–environment interactions shape microbial landscapes, this study provides a framework for targeted conservation strategies to protect climate-sensitive freshwater systems, emphasizing the role of macrophyte microbiomes in biogeochemical cycling and eutrophication mitigation. A critical constraint of this research stems from its exclusive focus on plant characteristics during a single growing season. Given that aquatic macrophytes exhibit marked variations in physiological traits across different phenological stages, subsequent studies should prioritize multi-seasonal sampling to capture this dynamic life-history modulation of phyllosphere microbiota assembly.

## Figures and Tables

**Figure 1 microorganisms-13-00960-f001:**
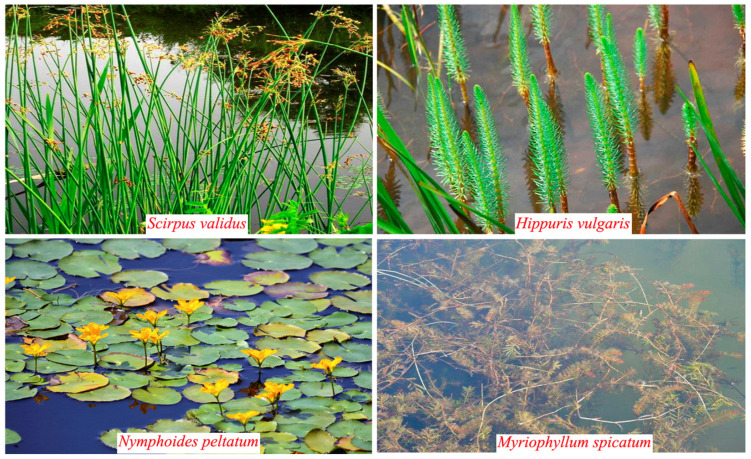
Four aquatic plants collected from the environment of Mayinghai Lake in Ningwu, Shanxi, China.

**Figure 2 microorganisms-13-00960-f002:**
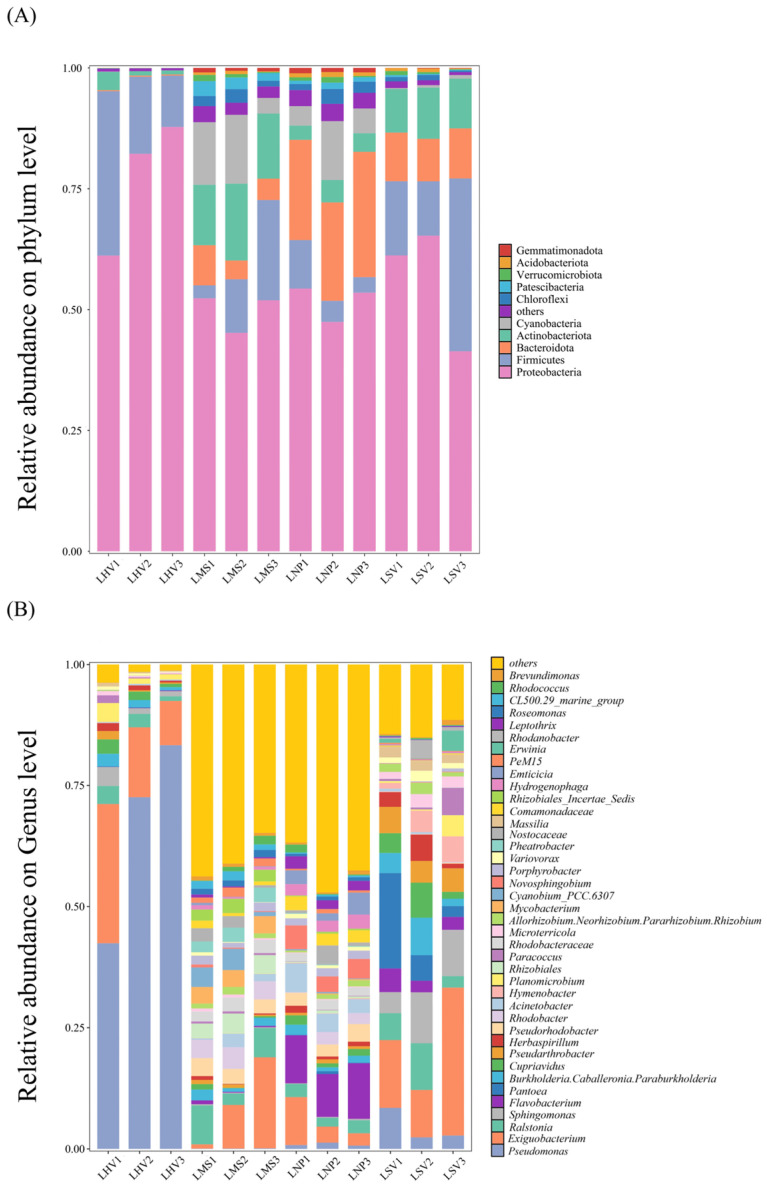
Compositions of epiphytic bacterial communities on the leaves of four aquatic plants: (**A**) dominant bacterial phyla; (**B**) dominant bacterial genera. LHV refers to the leaves of *Hippuris vulgaris*; LMS refers to the leaves of *Myriophyllum spicatum;* LNP refers to the leaves of *Nymphoides peltatum*; and LSV refers to the leaves of *Scirpus validus*. The same applies hereinafter.

**Figure 3 microorganisms-13-00960-f003:**
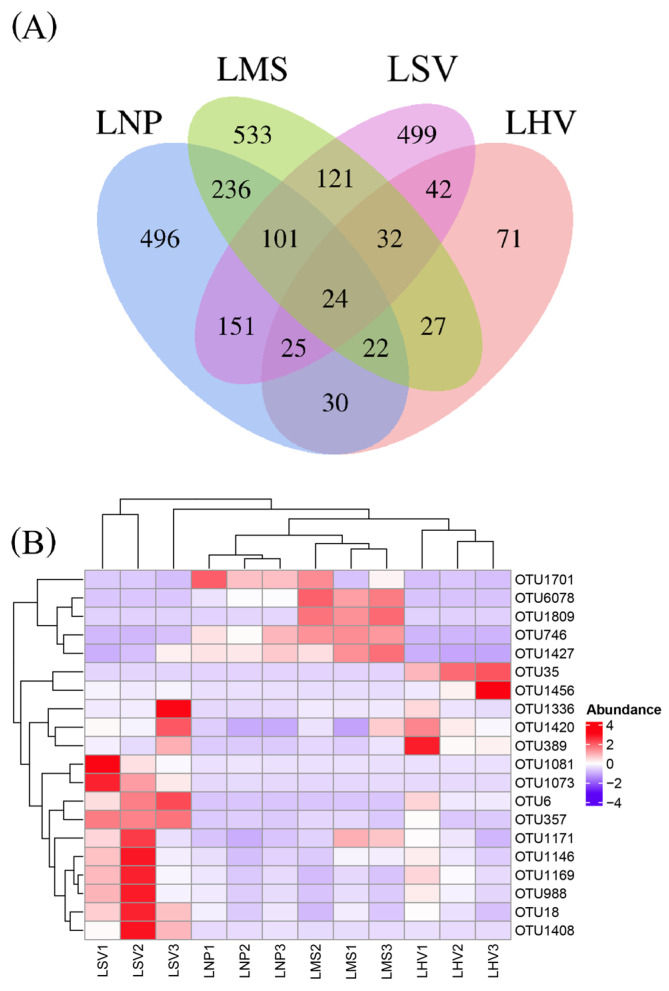
(**A**) Distribution characteristics of leaf-surface-associated bacterial communities of four aquatic plants at the OTU level and (**B**) heatmap of dominant shared OTUs distribution (relative abundance greater than 1%).

**Figure 4 microorganisms-13-00960-f004:**
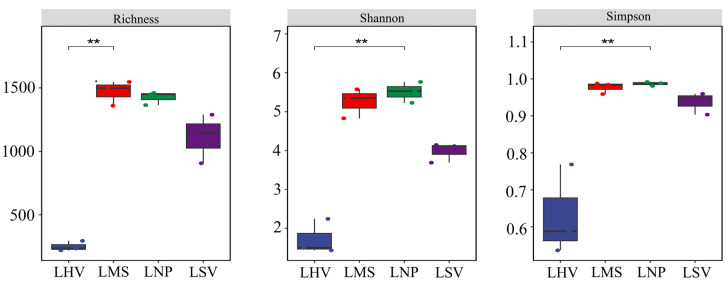
Alpha diversity indexes of the epiphytic bacterial communities. * Significance level, *p*: ** *p* < 0.01.

**Figure 5 microorganisms-13-00960-f005:**
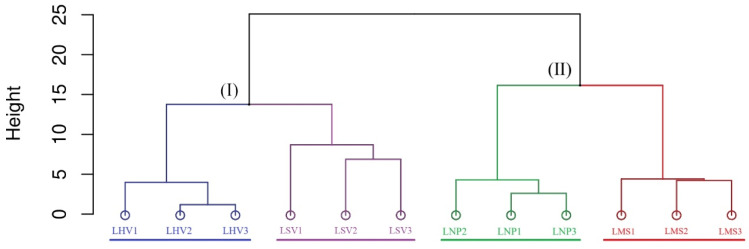
Hierarchical clustering analysis of epiphytic bacterial communities at the OTU level based on Bray–Curtis distance. Cluster I represents a group of data points with structurally similar properties (in terms of distances). Cluster II represents another group of data points that share similar internal structures.

**Figure 6 microorganisms-13-00960-f006:**
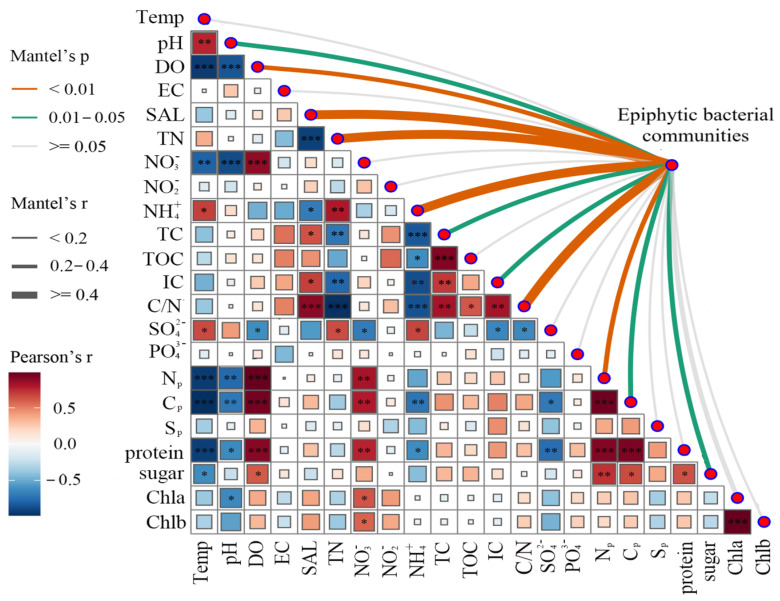
Mantel analysis of the plants’ epiphytic bacterial community. Edge widths in the network correspond to Mantel’s r values, and colors denote statistical significance. Edge width corresponds to the Mantel’s r statistic of distance correlation, while edge color denotes statistical significance. Np: nitrogen content in plant leaves; Cp: carbon content in plant leaves; Sp: sulfur content in plant leaves; Chla: chlorophyll a; Chlb: chlorophyll b. * Significance level, *p*: * *p* < 0.05, ** *p* < 0.01, *** *p* < 0.001.

**Figure 7 microorganisms-13-00960-f007:**
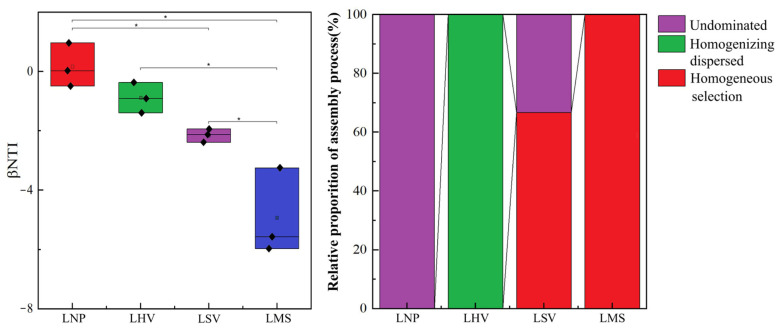
Construction process and βNTI values of the epiphytic bacterial communities of four aquatic plants. * Significance level, *p*: * *p* < 0.05. The diamond represents the value corresponding to the box plot.

**Figure 8 microorganisms-13-00960-f008:**
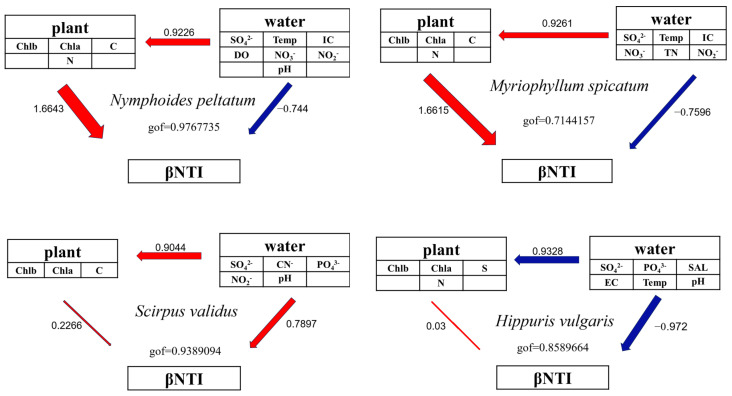
Path analysis of the epiphytic bacterial communities of four aquatic plants.

**Table 1 microorganisms-13-00960-t001:** The physicochemical parameters of water surrounding four types of aquatic plants.

Parameter	*Hippuris vulgaris*	*Myriophyllum spicatum*	*Nymphoides peltatum*	*Scirpus validus*
pH	6.89 ± 0.05 c	7.46 ± 0.17 a,b	6.89 ± 0.01 c	7.28 ± 0.05 b
DO (mg/L)	12.62 ± 0.04 b	9.87 ± 0.03 c	13.74 ± 0.05 a	12.15 ± 0.01 b
EC (μS/cm)	372.00 ± 83.14 a	393.33 ± 14.01 a	380.00 ± 31.24 a	512.67 ± 114.99 a
SAL (ng/L)	7.83 ± 0.21 a	5.27 ± 0.27 b	5.05 ± 1.39 b	7.94 ± 0.41 a
TN (mg/L)	0.91 ± 0.01 b,c	1.11 ± 0.02 b,c	1.13 ± 0.04 b	0.85 ± 0.01 c
${NO}_{3}^{-}$ (mg/L)	0.17 ± 0.00 a,b	0.13 ± 0.02 b	0.17 ± 0.00 a,b	0.15 ± 0.00 a,b
${NH}_{4}^{+}$ (mg/L)	0.55 ± 0.02 b	0.89 ± 0.03 a	0.61 ± 0.01 b	0.19 ± 0.03 c
TC (mg/L)	60.22 ± 0.03 b	59.31 ± 0.51 b,c	60.00 ± 1.07 b	62.67 ± 1.49 a
TOC (mg/L)	16.26 ± 0.30 b	16.08 ± 0.12 b	16.50 ± 1.38 a,b	17.94 ± 1.52 a
IC (mg/L)	43.96 ± 0.31 b	43.23 ± 0.40 c,d	43.50 ± 0.48 b,c	44.72 ± 0.13 a
C/N	66.18 ± 0.73 b	53.59 ± 1.06 c	53.01 ± 2.56 c	73.52 ± 2.74 a
${SO}_{4}^{2-}$ (mg/L)	45.36 ± 0.67 c	50.30 ± 2.76 a,b	47.20 ± 1.15 b	45.70 ± 0.26 c
${PO}_{4}^{3-}$ (mg/L)	0.26 ± 0.18 b	0.25 ± 0.03 b	0.28 ± 0.05 b	0.24 ± 0.05 b

Values are the mean ± standard deviation, and different lowercase letters represent significant differences (*p* < 0.05).

**Table 2 microorganisms-13-00960-t002:** The physiological indexes of the leaves of four aquatic plants.

Parameter	*Hippuris vulgaris*	*Myriophyllum spicatum*	*Nymphoides peltatum*	*Scirpus validus*
Leaf nitrogen content (mg/g)	2.98 ± 0.01 c	1.42 ± 0.00 d	4.10 ± 0.01 a	3.09 ± 0.03 b
Leaf carbon content (mg/g)	35.25 ± 0.07 c	20.96 ± 0.02 d	40.01 ± 0.16 a	37.62 ± 0.07 b
Leaf sulfur content (mg/g)	0.44 ± 0.02 a	0.40 ± 0.03 a	0.51 ± 0.25 a	0.55 ± 0.02 a
Soluble protein (mg/g)	7.77 ± 1.39 a	3.67 ± 0.98 b	9.29 ± 0.68 a	8.19 ± 0.88 a
Soluble sugar (mg/g)	2.77 ± 0.98 b	3.20 ± 0.27 b	8.99 ± 1.40 a	7.15 ± 0.75 a
Chlorophyll a (mg/g)	0.36 ± 0.05 a	0.16 ± 0.04 b	0.22 ± 0.10 b	0.17 ± 0.03 b
Chlorophyll b (mg/g)	2.98 ± 0.01 c	1.42 ± 0.00 d	0.11 ± 0.04 b	3.09 ± 0.03 b

Values are the mean ± standard deviation, and different lowercase letters represent significant differences (*p* < 0.05).

## Data Availability

The original contributions presented in this study are included in the article. Further inquiries can be directed to the corresponding author.

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
