# Peer review of "Host Lifeform Shapes Phyllospheric Microbiome Assembly in Mountain Lake: Deterministic Selection and Stochastic Colonization Dynamics"

_microorganisms, 2025, doi:10.3390/microorganisms13050960_

Round 1
Reviewer 1 Report
Comments and Suggestions for Authors
Comments for Authors
In this manuscript, the authors conducted a useful study with the aim of provide valuable insights on the Factors (biotic and abiotic) that most affect the assembly of epiphytic bacterial communities on aquatic macrophytes in a vulnerable subalpine lake ecosystem. The authors found a distinct microbial divergence among four macrophyte species. In particular, microbial β-diversity was found to be mainly influenced by water chemistry and plant functional characteristics. Overall, the authors showed a significant influence of plant life form in defining phyllosphere microbiomes, to the point of considering them a critical hierarchical filter structuring phyllosphere microbial landscapes.
The experimentation was well performed and the methods used are in accordance with analytical criteria.
The results exhibited and commented by the authors allow highlighting the ability of some aquatic macrophytes to facilitate the colonization specific bacterial colonies that play an essential role in macro- and micronutrient mineral cycles, promoting adaptation and resilience of autotrophic systems in the aquatic environment.
The work is original and can be accepted if the authors respond to the following reviewer's comments:
Since the analytical investigation was carried out correctly and linearly,the results cannot be questioned, so the few .comments will be only conceptual:
I would invite the authors to expand the introduction, perhaps by briefly comparing the aquatic microbiome with the soil microbiome, where in both cases plants play an important role. In my view, while in the biosphere, and particularly in the soil bioactive layer , there is a heterogeneous distribution of mineral nutrients and physicochemical parameters, the plant-microbiome relationship is more managed by environmental factors, so identical plants in different soils may provide a different bacterial assemblage and microbiome (see Di Martino et al 2022 Plants MDPI doi.org/10.3390/plants11070924).
In contrast, in the hydrosphere and particularly for lacustrine aquatic macrophytes, where there is a relative stability of chemical and physical factors (such as temperature, pH, and dissolved oxygen) and homogeneity in nutrient distribution, it is the biology and plant life form that define the microbiomes of the phyllosphere and structure the microbiological architecture of the aquatic ecosystem.
Line 49: The authors consider epiphytic microbial communities in a symbiotic relationship with the plant. I would not speak of a symbiotic relationship since epiphytic bacteria do not directly provide mineral nutrients to the plant, although they interact effectively in aquatic mineral cycles, so it would be more appropriate to speak of commensalism and inquinilism of the bacteria toward the host plant.
I ask the authors to make specific reference to the type of pollutants that epiphytic bacteria would break down levels to in a subalpine basin. I recall that mineralizable organic matter is not a pollutant, while polyphosphates from industrially synthesized detergents and a cause of eutrophication may be considered as such, but I wonder how they could end up in a natural subalpine basin.
I ask the authors to clarify whether the mentioned macrophytes suffer biological competition, that is, whether the presence of one can harm the other or whether they coexist in a biostable relationship.
Finally, I invite the authors to indicate among the identified epiphytic bacteria which ones are most involved in the mineral cycle of nitrogen and phosphorus, which are important macronutrients of plants, and whether these bacteria are tolerant to moderate salinity conditions with the assumption that they can transfer and employ them in soils subject to moderate salinity.
Reviewer 2 Report
Comments and Suggestions for Authors
Microorganisms
Manuscript Draft
Manuscript Number: 3579220
Title: Host Lifeform Shapes Phyllospheric Microbiome Assembly in Mountain Lake: Deterministic Selection and Stochastic Colonization Dynamics
Article Type: Research article
General Comments on MDPI Questions that Reviewers must answer:
- Is the manuscript clear, relevant for the field and presented in a well-structured manner?
This manuscript is written clearly, is very well-structured, and is potentially relevant to the field since it conducted analyses of bacterial communities on four plant species in a subalpine lake in northeastern China. Given the potential contribution of this research, this manuscript has potential but requires more improvement to warrant publication in MDPI Microorganisms. Please make the following SIX general edits:
1) In the last paragraph of the Introduction section, please more clearly state the goal(s) and then the objective(s) of the research. Generally there is a big overall goal. Please state this goal. Then outline the objective(s) of the research not as research questions but what was set out to do in terms of objective(s). Generally there is more than one objective of the research. Please number these as you have done your research questions.
2) Please add a few more paragraphs to the Introduction in terms of the context needed for the reader to better understand your research. The topics should be regarding background information on the four plant species studied as well as the bacterial and microbial communities associated with these types of plants and why this is important. You will need to cite more literature to support the new writing. The Introduction section as currently written is too short.
3) Please add a new Figure 1 in the Materials and Methods section showing these four aquatic plant species in their subalpine lake environment.
4) At the start of each major section of the manuscript as well as in Tables, please at first use of the Genus species please write out the Genus in full (then abbreviate as afterward).
5) In the Discussion section, please break this up into two subsections 4.1. Contrasts to Past Research and 4.2. Research Implications. This will make it clear to the reader how the results compare to past studies. Most of what is currently written would be 4.1.1. and 4.1.2. and 4.1.3. under 4.1. Contrasts to Past Research. Please write the newly added sub-section 4.2. Research Implications on the important implications of the research results (note in the Abstract you have written “These findings establish host lifeform as a critical hierarchical filter structuring phyllospheric microbial landscapes, with cascading implications for developing targeted conservation strategies in climate-sensitive highland lacustrine systems.” and this was not discussed at all). What exactly are the cascading implications for targeted conservation? What is being conserved? What do the research results have to do with climate-sensitive highland lacustrine systems? Do your research results have implications for mitigating adverse impacts related to what you are implying? Please expand on this and cite more literature to support your discussion.
6) Please refer back to the MDPI Microorganisms Word template for authors on the journal website and make sure all the citations in the References section follow the exact formatting. Please also add DOI links for all references at the end of the citation.
Please also make the following EIGHT minor edits and clarifications:
1) The summary in the left margin toward the bottom of the first page is missing and needs to be added back in
2) On L31-32, the keywords need to be in alphabetical order and the words should not be capitalized
3) Paragraphs by definition have a minimum of 3 sentences (1 topic sentence followed by a minimum of 2 supporting sentences). For example on L190-193, there are only 2 sentences. Please correct elsewhere in the manuscript by either adding sentence(s) or merging such writing with another paragraph
4) Everywhere in the manuscript, please change Fig. to Figure
5) For in-text citations, please use the longer endash symbol and not the shorter hyphen and the endash is the same symbol used for page number ranges so for example on L277 please use the endash and not the hyphen as please do this elsewhere if warranted
6) In Table 1, please add a blank space between all numbers +/- and the letter used to denote statistically significant differences across each row so that these results are more visually understandable
7) For Figure 2, put (B) under (A) and make (B) bigger so the writing can be read (you may have to also increase font size)
8) On L139, change to Results
- Are the cited references mostly recent publications (within the last 5 years) and relevant? Does it include an excessive number of self-citations?
Only 9 of the 32 cited references have been published within the last 5 years since 2019. Please cite more recent literature and please increase the number of citations. All citations appear relevant to the research topic and there are no excessive self-citations (please make sure this is still the case with citations that you will add).
- Is the manuscript scientifically sound and is the experimental design appropriate to test the hypothesis?
The manuscript is scientifically sound and the analyses of bacterial species assessment are comprehensive.
- Are the manuscript’s results reproducible based on the details given in the methods section?
The manuscript’s results are reproducible after reading the 2. Materials and Methods section.
- Are the figures/tables/images/schemes appropriate? Do they properly show the data? Are they easy to interpret and understand? Is the data interpreted appropriately and consistently throughout the manuscript? Please include details regarding the statistical analysis or data acquired from specific databases.
Please make edits to figures and tables that were already discussed.
- Are the conclusions consistent with the evidence and arguments presented?
The Conclusions appear to be consistent with the evidence and arguments presented. Please add a couple of sentences to the end of the paragraph in the 5. Conclusions section on how future research can improve upon your results.
- Please evaluate the data availability statements to ensure it is adequate.
The Back Matter sections between the Conclusions and the References need to follow the exact formatting of the MDPI Microorganisms Word template for authors on the website. For example, co-author initials need to be used and not full names for Author Contributions: and there are missing sections including the data availability statement and ethics statement.
Reviewer 3 Report
Comments and Suggestions for Authors
The research objectives of this manuscript are clearly formulated. This manuscript presents the composition and diversity of epiphytic bacterial communities on leaves of four macrophyte aquatic species: Scirpus validus, Hippuris vulgaris, Nymphoides peltatum, and Myriophyllum spicatum, which represent different life forms (emergent, floating-leaved, submerged). Also investigated were: i) water chemistry parameters (nitrate, nitrite, pH) and ii) plant leaves traits (chlorophyll, nitrogen, carbon content). Based on High-Throughput Sequencing, 3,580 operational taxonomic units (OTUs) were obtained. Various analyses enabled the authors to determine which factors have a fundamental influence on the formation of bacterial communities and their alpha and beta diversity in the phyllosphere of aquqtic macrophytes. The obtained results are very valuable and bring new aspects to science. Most of them are presented in a clear and transparent manner. However, there are some errors and some discrepancies between the text in Results and Discussion (see Remarks). It is advisable to delete from the Discussion section the texts that are a repetition of the results. After minor revision, the manuscript should be published in Microorganisms.
Remarks
Line 22-23 this sentence needs correction
Line 59 taxa[6] – add a space
Line 150 there are numerous errors in Table 1 that require correction PO3- 4, NH+ 4, NO–3, SO2- 4. These data must be consistent with the text.
Line 157 species.At - add a space
Line 163- 166 Pseudomonas, Exiguobacterium, Flavobacterium – it should be in italic. This should also be corrected in other places in the manuscript
Line 168-171 the text should be corrected
Line 168 In Figure 1 there is no explanation of which phyla are represented by the individual colors
Line 172-175 this text should be deleted as it is a repetition
Line 206 (Fig.5) - add a space
Line 235 minor corrections should be made
Line 236-237 this text is inconsistent with the text given in Results in lines 160-161. This requires correction.
Line 239 α-Proteobacteria - this term appears in the Abstract, Discussion and Conclusion. However, it is missing in the Results. This requires supplementation.
Line 240 this text is not consistent with the text on lines 160-161 – this requires explanation
Line 241 g__Mycobacterium - it is unclear what this term is supposed to mean
Line 260 beta diversity - why is this term not used at all in Results (Fig.4, Fig. 5) but only in Discussion?
Line 291 γ-Proteobacteria - this term appears in Discussion and Conclusion. This requires supplementation, especially in the Results section.
Line 291 H. vulgaris (submerged), please note that in Material and Methods in line 82 H. vulgaris has been classified as emergent plants. This should be checked in the entire manuscript and corrected.
Line 381 Zhang, here the font is different than in numbers 1-31.
Reviewer 4 Report
Comments and Suggestions for Authors
The authors present an interesting study on the microbial diversity analysis of microbial communities of subalpine aquatic plants and drivers of the composition of their communities. The authors provide sufficient introductions, and the methods are properly described. However, I lack information about the volume of samples collected and the depth of the water samples. The authors have analyzed the community based on 16S DNA sequence, which is a standard procedure but is methodically burdened. This is because bacteria differ in the number of copies of the 16S rRNA gene. Therefore, the abundance of 16S DNA does not directly translate to the number of bacterial cells. However, as long as you compare the results obtained with the same methodology, this could be used to obtain meaningful results, but please bear that in mind when comparing the abundance of different OTUs. I also miss the table with the detected otu abundance used to prepare Figure 1 as it is hard to read values and analyze the abundance of different OTUs from this figure. This proposed table could be added as a supplementary file. The authors could also include some photographs of analyzed plants and maybe use their schematic representations for the prepared graphics, which would help in the interpretation of the presented figures. Concluding, I recommend this article to be accepted for publication in Microorganisms after minor revisions.
Line 210: Please explain the dots inside the squares
Line 228: Please add names of the plants to this graphic
Back matter lacks some sections: eg. Conflict of interest, etc. Please refer to the instructions for authors for the journal
The literature list has all names in all caps, while only the first letters should be capitalized. Please refer to the journal literature formatting instructions for authors.
Reviewer 5 Report
Comments and Suggestions for Authors
Review of the Manuscript: "Host Lifeform Shapes Phyllospheric Microbiome Assembly in Mountain Lake: Deterministic Selection and Stochastic Colonization Dynamics"
Manuscript ID: microorganisms-3579220-peer-review-v1
Dear Authors,
I have reviewed the manuscript and recognize its potential relevance to the field of microbial ecology, particularly in the context of aquatic ecosystems. The combination of 16S rRNA sequencing, community assembly modeling (βNTI), and structural equation modeling represents a robust approach.
However, the manuscript currently has several critical weaknesses that must be addressed to meet publication standards. I recommend major revisions before reconsideration. The selection of the four macrophyte species is not sufficiently justified, and details about replication to avoid pseudoreplication are missing. Environmental data are limited, with important factors such as water movement, plant age, and leaf surface characteristics being omitted, although they can significantly influence microbiome composition. Sampling was conducted at a single time point (August 2020) without seasonal replication, which weakens the robustness of the conclusions. The discussion on stochasticity and determinism is too general, and the interpretation of βNTI requires additional support. Some sections are vague and should be written more clearly, and figures need improvements, such as indicating the percentage of variance explained.
Recommendation: Reconsider after major revisions.
Best regards,

Round 2
Reviewer 1 Report
Comments and Suggestions for Authors
The authors responded to the reviewers' comments thoroughly and convincingly, making changes and improvements to the manuscript with appropriate and direct language.
The manuscript can be accepted in its current form.
Reviewer 2 Report
Comments and Suggestions for Authors
Thank you for making requested edits.
Reviewer 5 Report
Comments and Suggestions for Authors
Journal: Microorganisms (ISSN 2076-2607)
Manuscript: IDmicroorganisms-3579220
Type: Article
Title: Host Lifeform Shapes Phyllospheric Microbiome Assembly in Mountain Lake: Deterministic Selection and Stochastic Colonization Dynamics
Dear Aurors,
Congratulations to the authors for their dedicated work. It is clear that they have thoughtfully addressed the previous comments and implemented the suggested changes with care and precision. The manuscript has improved significantly in clarity, structure, and scientific depth. I particularly appreciate the responsiveness to feedback, which has clearly elevated the quality of the work. In its current form, I have no further recommendations the manuscript is complete, compelling, and I am more than pleased to recommend it for publication.
Best regrds.
